# Harnessing Ionic Interactions and Hydrogen Bonding for Nucleophilic Fluorination

**DOI:** 10.3390/molecules25030721

**Published:** 2020-02-07

**Authors:** Young-Ho Oh, Hyoju Choi, Chanho Park, Dong Wook Kim, Sungyul Lee

**Affiliations:** 1Department of Applied Chemistry, Kyung Hee University, Gyeonggi 17104, Korea; chem_yhoh@daum.net (Y.-H.O.); gywn78@naver.com (H.C.); 2Department of Chemistry and Chemical Engineering, Inha University, Incheon 402-751, Korea; qazwse1112@naver.com

**Keywords:** ionic interactions, hydrogen bonding, nucleophilic fluorination

## Abstract

We review recent works for nucleophilic fluorination of organic compounds in which the Coulombic interactions between ionic species and/or hydrogen bonding affect the outcome of the reaction. S_N_2 fluorination of aliphatic compounds promoted by ionic liquids is first discussed, focusing on the mechanistic features for reaction using alkali metal fluorides. The influence of the interplay of ionic liquid cation, anion, nucleophile and counter-cation is treated in detail. The role of ionic liquid as bifunctional (both electrophilic and nucleophilic) activator is envisaged. We also review the S_N_Ar fluorination of diaryliodonium salts from the same perspective. Nucleophilic fluorination of guanidine-containing of diaryliodonium salts, which are capable of forming hydrogen bonds with the nucleophile, is exemplified as an excellent case where ionic interactions and hydrogen bonding significantly affect the efficiency of reaction. The origin of experimental observation for the strong dependence of fluorination yields on the positions of -Boc protection is understood in terms of the location of the nucleophile with respect to the reaction center, being either close to far from it. Recent advances in the synthesis of [^18^F]F-dopa are also cited in relation to S_N_Ar fluorination of diaryliodonium salts. Discussions are made with a focus on tailor-making promoters and solvent engineering based on ionic interactions and hydrogen bonding.

## 1. Introduction

Nucleophilic fluorination [1,2,3,4] using various sources of F fluoride exhibits several advantages over the electrophilic [5,6] counterpart, especially for introducing the isotopic F-fluorine onto organic compounds: First, it does not need to use the carrier added [^18^F]F_2_ gas that are very cumbersome to handle. This feature is especially important when the radioisotopic fluorine-18 [^18^F] is to be incorporated into an organic substance for clinical applications to positron emission tomography (PET) [7,8,9]. Second, [^18^F]F-labeled molecules can generally be produced in higher radiochemical yield and higher activity than the electrophilic substitution using carrier added [^18^F]F_2_ sources by several orders of magnitude. Third, the reaction yields from nucleophilic fluorination may improve tremendously (up to > 90%) in reasonable reaction time (<2 h) by employing a variety of promoter/catalyst with a minimum amount of by-products. Recent use of alkali metal fluoride in ionic liquids (ILs) have clearly opened this possibility by designing and applying various ‘task-specific’ ILs for nucleophilic fluorination. This novel capability of ILs as promoter/catalyst is really the results of the fact that ILs comprise the ionic species (cation and anion), but the detailed mechanism has been seldom understood. Here we review recent advances in the catalysis/promotion of S_N_2 fluorination by ILs, focusing on the mechanistic features of the process. We show that the rates and yields of nucleophilic fluorination may be improved by monitoring and controlling the Coulombic forces and weak interactions (hydrogen bonding, π-interactions, etc.) between IL cation, anion and substrates (especially, the leaving group).

These interactions also determine the efficiency of S_N_Ar fluorination of diaryliodonium salts that are gaining much importance as a useful path to incorporating ^18^F and ^19^F to aromatic compounds. Since diaryliodonium salts consist of diaryliodonium cation and the counter-anion such as Br^−^, OTf^−^ etc., Coulombic interactions with the alkali metal counter-cation and the nucleophile F^−^ would be critical to determine the efficiency of fluorination. When the substrate contains functional groups such as hydroxyl (-OH), amino (-NH_2_, -NHR) that are amenable to hydrogen bonding, the situation becomes too complicated to scrutinize by intuition only. In the second part of this brief review, we show that nucleophilic fluorinations of diaryliodonium salts using alkali metal fluoride are indeed significantly affected by these electrostatic interactions. In some cases, the position of the nucleophile F^−^ relative to those of the electropositive C atom and the leaving group may be determined by these ionic interactions and hydrogen bonding. Either the nucleophile F^−^ is positioned close to the electropositive C atom for efficient fluorination, or F^−^ may be located far from the center of reaction as the results of intricate interplay of the electrostatic interactions. In the latter case, of course, fluorination will not proceed at all. In order to scrutinize the configuration of the reacting system (diaryliodonium cation, counter-anion, leaving group, metal cation and F^−^) in pre-reaction complex and transition state to analyze the experimentally observed efficiency of reaction, quantum chemical calculations are indispensable, and we discuss the relevant recent works.

## 2. S_N_2 Fluorination in Ionic Liquids

Besides being considered excellent solvent for chemical reactions because of its many useful physicochemical properties such as very low vapor pressure, non-combustibility, high thermal stability, low viscosity, easy recovery and high ionic conductivity, ionic liquids (ILs) [10,11,12,13,14,15,16,17,18,19] have found further significant role as catalysts/promoters [20,21] in many chemical transformations such as S_N_2 [22,23,24,25,26,27,28], Diels-Alder [29], aldol condensation [30], Heck [31,32,33], and Michael addition [34] reactions. The acceleration of reaction rates of organic reactions by IL occur by the nature of the substance comprising cation and anion. However, the detailed mechanism seldom seems to be fully elucidated. Song and coworkers [26] gave a detailed review for experimental observations of increasing catalytic activity of metal triflates such as Sc(OTf)_3_ in ILs, attributing the phenomenon to formation of superacidic Lewis acidic catalysts by anion exchange. They systematically studied the “anion effect” for reactions such as Diels-Alder, and Friedel-Crafts Alkylation (Scheme 1). Therefore, we skip the discussion on this topic, and only review the catalytic activity of ILs in organic solvents or in solvent-free environment. 

The first demonstration (Table 1) of promotion of nucleophilic fluorination in ionic liquids reaction media was made by Kim, Song and Chi [23,35]. They reported that KF is the source of F^−^ in imidazolium based ILs [bmim][X] resulted in excellent yields (>90%) in reasonable reaction time (<2 h) with a minimal amount of by-products. Use of cosolvent such as acetonitrile did not affect or improve the reactivity of fluorination. They also showed that fluorination in [bmim][X] did not require anhydrous condition, demonstrating the extreme flexibility of ionic liquids for reaction conditions. However, less than stoichiometric amounts of IL (0.5 equiv) as catalyst/promoter showed rather poor performance for the nucleophilic fluorination.

The mechanism of these very interesting observations was elucidated by Lee and co-workers [24] in quantum chemical analysis. It was found that the ionic liquid anion plays a key role as a Lewis base by binding to the counter-cation K^+^ or Cs^+^, thereby reducing its retarding Coulombic influence on the nucleophile F^−^. That is, the counter-cation becomes a ‘ghost-like’ agent by the action of the negative charge of IL anion. The intricate interplay (Coulombic forces) of the counter-cation, the nucleophile, IL cation and anion helps to form a pre-reaction complex and transition state that is optimal for S_N_2 fluorination, prohibiting the formation of by-products (Figure 1). The role of IL anion in this mechanism corresponds to that of electronegative O atoms in oligoethylene glycols [36] or in bulky alcohols [37,38,39] (*t*-butanol and amyl alcohol) that were proved to act as catalyst/promoter in nucleophilic fluorination, but were probably better because of the explicit negative charge of IL anions. 

Magnier and coworkers [40] carried out a somewhat different approach to using ILs for nucleophilic fluorination. Unlike the experiments listed in Table 1, in which the source of the fluorinating agent is metal fluoride, they employed IL [bmim][F] directly, thereby in solvent-free environment. The IL [bmim][F] was prepared by the exchange of [bmim][Cl] with KF. Another interesting observation was that the reaction yields increased from 49 to 84 and 95% as the equivalent of IL increased from 1:1 to 1:2 and 1:3 (Table 2). The underlying mechanism [41] of Magnier and coworkers observations is depicted in Figure 2. When 1 eq. of IL is employed, the calculated pre-dissociation complex is shown in Figure 2a. Here, the IL cation bmim^+^ bridges the nucleophile F^−^ and the leaving group -OTs, helping the formation of compact pre-reaction complex with optimal configuration for S_N_2 fluorination. For 2 eq. of [bmim][F] used (Figure 2b), the role of F^−^ is two- fold: One of the two F^−^ acts as a regular nucleophile, whereas the other one is off the electropositive carbon and the leaving group, whose action is on the two IL bmim^+^ cations to reduce their strong Coulombic forces on the nucleophile. 

It seems that Magnier and coworkers’ experiments were the first implicit demonstration of the *contact ion-pair* S_N_2 mechanism proposed by Lee and co-workers [42], in that the separation of IL cation and anion in liquid phase [bmim][F] would be extremely difficult in reaction medium because of the strong electrostatic force between them. The calculated pre-reaction complexes (whose coordinates are given in Appendix A) shown in Figure 2 also support this suggestion that IL counter-cation and anion/nucleophile are in close contact. This role of IL for promoting the process of nucleophilic fluorination may also be described as an electrophile – nucleophile dual activator proposed by Lu and co-workers [43].

As a “task-specific” promoter/catalyst for organic transformations to be designed, the structure of side-chain [44] of IL cation and anion would be an important determinant of the efficiency of ILs. Modification of ILs by oligoethylene glycol chain is a good example, in which the synergistic effects of the two moieties improve the overall efficiency of the promoter/catalyst (Figure 3) [24].

Another example is the design and use of pyrene-tagged IL (Figure 4) [45]. As presented in Figure 4, the pyrene moiety acts as an additional Lewis base besides the IL anion in this mechanism, further strengthening the catalytic efficiency of IL. The mechanism (Figure 5) (pre-reaction complex → transition state → post-reaction complex) of the reaction is depicted in Figure 5: Pyrene - cation π interactions facilitate the reaction by giving π electrons to the counter-cation, further alleviating the Coulombic influence of the cation on the nucleophile [24]. 

## 3. S_N_Ar Fluorination of Diaryliodonium Salts

As in the case of S_N_2 fluorination, electrophilic S_N_Ar fluorination such as fluorodestannylation [46] using [^18^F]F_2_ suffers similar kinds of drawbacks. A conventional approach to incorporating F to aromatic compounds was to try nucleophilic fluorination of electron deficient aromatics. Use of diaryliodonium salts have now proved very useful for the preparation of [^18^F]fluoroaromatics electron-rich rings at any desired ring position. This methodology is particularly important in relation to the synthesis of many useful [^18^F] radiophamaceutical compounds to be clinically detected by non-evasive PET technique. The seminal work by Pike and coworkers [47] was published in 1998, and since then this approach has found wide applicability [48,49,50,51,52,53]. This feature of diaryliodonium salts seems to originate from the ionic nature of the compounds. There exists an excellent review by Gouverneur and coworkers, [8] so we will not try to be comprehensive in this concise review. In some studies, metals such as copper [54,55,56] were employed to carry out S_N_Ar fluorination of diaryliodonium salts, but we will also leave this topic to other reviews. We will only cite three typical studies: Coenen and coworkers’ [49] metal-free S_N_Ar fluorination of diaryliodonium salts containing the 2-thienyl group (Scheme 2) is a good example of regioselective no-carrier-added [49,57,58] radiofluorination. The influence of the substitution pattern, of counteranions, and of different reaction conditions were studied carefully. Seid et al. studied one-pot synthesis of unsymmetrical aryl(2,4,6-trimethoxyphenyl)iodonium salts [57]. Chun and coworkers’ [58] chemoselective radiosynthesis of [^18^F]fluoroarenes using an aryl(2,4,6-trimethoxyphenyl)iodonium tosylate for ^18^F-incorporation on electron-rich aryl rings (Scheme 3) is a most recent study.

S_N_Ar fluorination of diaryliodonium salts may even give products that are contrary to what can be expected by conventional inductive effects: [^18^F] labeling may occur at the ring with larger electron density with RCY up to 68% in 30 min. (Table 3) [59].

When aromatic fluorination is carried out for diaryliodonium salts with alkali metal fluoride, complicated ionic interactions would arise among the diaryliodonium cation, counter-anion, metal cation and the nuclephile F^−^, and they would certainly affect the outcomes of the reaction significantly. Presence of ring substituents and/or the leaving group prone to forming hydrogen bonding with ionic species would cause further complications. These interactions may make the elucidation of the mechanism by intuition a formidable task. However, they may constitute a very instructive approach to synthesizing [^18^F]fluoroaromatics that are difficult to achieve by conventional methods. 

One such strategy results from the question: Could we monitor and control the position of F^−^ relative to the S_N_Ar center (that is, the electropositive C atom at which the S_N_Ar occur) by harnessing these interactions? Among the numerous studies of S_N_Ar fluorination, diaryliodonium salts containing side chains that are amenable to form hydrogen bonds seem to be the best example to address this question.

One such example is the aromatic ^18^F-labeling of guanidine-containing radiopharmaceutical [60,61,62,63,64,65,66,67,68,69,70] that was synthesized by nucleophilic fluorination of diaryliodonium salts. Jang and coworkers [71] found that the positions of -Boc protection profoundly affected the efficiency of ^18^F-labeling: The fully protected N,N’,N¨,N¨-tetrakis-Boc guanidine group (1b in Scheme 4) exhibited remarkably enhanced reactivity (yield = 39% in 5 min) and improved selectivity in contrast to N,N’-bis-Boc protected guanidine (1a, yield ≈ 0) in the absence of hydrogen bonding with fluoride ion (Scheme 4). What is the origin of these very intriguing observations? The observed difference in reactivity of (1a) and (1b) may be understood by scrutinizing the structures of corresponding pre-reaction complexes depicted in Figure 6: It was proposed by quantum chemical analysis that strong Coulombic interactions among the ionic species and hydrogen bonding between the anion and the amino group –NHR resulted in very different position of the nucleophile relative to the electropositive C atom. As shown in Figure 6, the nucleophile F^−^ in (pre-8a) forms a hydrogen bond with the amino group in guanidine, and this interaction moves the nucleophile far away from the iodonium site. Thus, the S_N_Ar reaction may never proceed from this complex. On the other hand, in (pre-8b), the absence of the hydrogen bonding with the fully protected (N,N’,N¨,N¨-tetrakis-Boc) amine group allows F^−^ to be near the iodonium site, and therefore from this favorable configuration the nuclephilic attack of F^−^ onto the electropositive C atom may easily ensue. This example seems to give a good answer to the question posed above: By carefully monitoring the ionic interactions and hydrogen bonding, it is possible to locate the nucleophile at a position that is favorable for S_N_Ar reactions.

Another ^18^F-labeled radiopharmaceutical that is under intense investigation recently by S_N_Ar fluorination is [^18^F]F-dopa. There have been considerable amount of efforts [72,73,74,75,76] to prepare this radiotracer over the years. Neumaier and coworkers work based on Ritter and co-workers’ [74] scheme (Scheme 5) of nickel-based fluoride-derived electrophilic radiofluorination reagent is a good example. Numerous attempts to synthesize [^18^F]F-dopa by S_N_Ar scheme have also experienced severe problems of poor yields along with many undesirable by-products. Besides these problems, the number of steps in the synthesis of this elusive radiotracer is of profound importance in order to be used for clinical purposes. Wirth and co-workers’ recent work [72] seems to deserve attention because of the simplicity of the synthetic scheme (Scheme 6) (S_N_Ar fluorination of diaryl iodonium salt based on DiMagno and co-workers procedure [49]) despite the low yields (0–5%) of both “cold” (^19^F) and “hot” ([^18^F]) fluorination. The origin of the observed poor yields was also studied by quantum chemical calculations [75]: Focusing on the counter-anion Br^−^ in diaryliodonium salt, it was proposed that S_N_Ar bromination (Model I in Scheme 7) would be the predominant process rather than fluorination, resulting in very small amount of fluorinated product. This suggests that ionic interactions with the counter-anion (Br^−^ in Wirth’s experiments) strongly affect the reaction yield, and using a counter-cation such as OTf^−^ with lower nucleophilicity seems to be preferable. In ‘hot’ fluorination for synthesizing [^18^F]F-dopa, further complication would result from the use of base such as K_2_CO_3_ employed to extract ^18^F^−^. The alkali metal cation K^+^ may certainly be unfavorable due to its interactions with the nucleophile ^18^F^−^, and using Cs^+^ instead may be better. More recent experimental work by Maisonial-Besset et al. [76] for [^18^F] labelling of L-dopa (Scheme 8) without the use of base, cryptand or metal catalyst would also be of interest, because their methods gave improved RCY (27–38%) for [^18^F] labeling. It would be an excellent system to be investigated by quantum chemical methods. It seems that careful choice of the counter-anion (to the diaryiodonium) and counter-cation (to the nucleophile) and the base may lead to robust synthetic route to [^18^F]F-dopa with excellent yield and in short reaction time sufficiently satisfactory for clinical usage.

## 4. Conclusions

Nucleophilic fluorination, and especially [^18^F]Fluorination, is usually very difficult due to unfavorable solvent effects on the small-sized F fluoride when the reaction is attempted in organic solvent. Careful choice of efficient promoter/catalyst is usually required for this process. Use of alkali metal fluoride in Lewis base solvent/promoter such as bulky alcohols and oligoethylene glycols proved to be such a breakthrough for efficient nucleophilic fluorination. Employing ILs for that purpose is another example, in which the IL anion acts as Lewis base due to its ionic nature. We discussed that systematic design of the structure of ILs allowed very efficient S_N_2 fluorination with a minimum amount of by-products in reasonable reaction time. The harnessing of ionic interactions and hydrogen bonding also proved critical in controlling the reaction yield and rates. For S_N_Ar fluorination of diaryliodonium salts, we presented a number of examples where quantum chemical analysis allowed the elucidation of underlying mechanism and design of experiments. We presented the synthesis of [^18^F]F-dopa by S_N_Ar fluorination of diaryliodonium salts as a very recent example. We believe that our present review would help to advance further developments concerning the tailor-making of promoters/catalysts and solvent engineering based on ionic interactions and hydrogen bonding in the field of organic synthesis, medicinal chemistry, and environmental chemistry.

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
