# Peer review of "Harnessing Ionic Interactions and Hydrogen Bonding for Nucleophilic Fluorination"

_molecules, 2020, doi:10.3390/molecules25030721_

Round 1
Reviewer 1 Report
This MS provides a complete overview of recent works for the nucleophilic fluorination of organic compounds by Coulombic interactions and/or hydrogen bonding. This review is well organized and interesting for readers of Molecules.
I recommend its publication after the following minor revisions:
- Abstract and Conclusions paragraphs should be revised in order to highlight the future perspective of nucleophilic fluorination of organic compounds.
- Figs. 1,2,5,6. The front size should be increased to improve the readability of the figures.
Author Response
Abstract and Conclusions paragraphs should be revised in order to highlight the future perspective of nucleophilic fluorination of organic compounds.We add the following paragraph at the end of Abstract:
Discussions are made with focus on tailor-making of promoters and solvent engineering based on ionic interactions and hydrogen bonding.
We revise the last paragraph of Conclusion to:
We believe that our present review would help to advance further developments concerning the tailor-making of promoters/catalysts and solvent engineering based on ionic interactions and hydrogen bonding in the field of organic synthesis, medicinal chemistry, and environmental chemistry.
2. Figs. 1,2,5,6. The front size should be increased to improve the readability of the figures.
Font size is increased in Figs. 1,2,5,6.
Reviewer 2 Report
The authors review recent works for nucleophilic fluorination of organic compounds in which the Coulombic interactions between ionic species and/or hydrogen bonding affect the outcome of the reaction. The topic is highly interesting and could be of real interest to the scientific community, however some parts need to be addressed and some major revisions need to be undertaken. The review starts well but some parts at the end get quite confusing. I mainly recommend a major restructuring of the 3rd part (SNAr fluorination of diaryliodonium salts), along with some other minor comments.
After these comments have been addressed, the paper may be ready for publication
Add to the introduction, the review by R. L. Vekariya (Journal of Molecular Liquids 2017, 227, 44-60) Add CO2 application to the introduction (see mini review by He and coworkers in Frontiers in chemistry, titled: Ionic Liquids Catalysis for Carbon Dioxide Conversion With Nucleophiles) The numbering of substrates and products is lacking significantly.. what is the structure of 1? Illustrate please the structures in Table 3. And why does it jump to 8a and 8b afterwards? The L-dopa part is particularly full of inconsistencies and full of typos and grammar errors which make it very difficult to read. Scheme 5 is incomplete. Reaction conditions on arrow are missing. Also scheme looks copy pasted from somewhere. This needs to be redone. Triboc should be tri-Boc Scheme 7 needs to be reformatted properly Reagents in scheme 8 are missing.All in all the review starts well but when it gets to the SNAr fluorination of diaryliodonium salts part, the review becomes disconnected and inconsistent as if it was put together in a rush. I suggest that this part be reworked completely with more efforts toward format and organisation of data.
Author Response
1. Add to the introduction, the review by R. L. Vekariya (Journal of Molecular Liquids 2017, 227, 44-60).
We cited the review by R. L. Vekariya at the 4th line (# 67) in Section <SN2 fluorination in ionic liquids> as Ref. 20.
2. Add CO2 application to the introduction (see mini review by He and coworkers in Frontiers in chemistry, titled: Ionic Liquids Catalysis for Carbon Dioxide Conversion With Nucleophiles)
We cited the mini review by He and coworkers at the 4th line (# 67) in Section <SN2 fluorination in ionic liquids> as Ref. 21.
3. The numbering of substrates and products is lacking significantly. What is the structure of 1? Illustrate please the structures in Table 3. And why does it jump to 8a and 8b afterwards?
Substrates (1a) – (1f) are added to table 3. In Scheme 4 and Figure 6, the numberings of the compounds are revised from (8a) and (8-b) to (1a) and (1b), respectively.
4. Scheme 5 is incomplete. Reaction conditions on arrow are missing. Also scheme looks copy pasted from somewhere. This needs to be redone. Triboc should be tri-Boc
We added the reaction conditions (acetone, 85 °C, 10 min.) above the arrow. Scheme 5 was redone from Ref. 73. Triboc is revised to tri-Boc in Scheme 6.
5. Scheme 7 needs to be reformatted properly.
Scheme 7 is redone.
6. Reagents in scheme 8 are missing.
Reaction conditions and reagents are added above the first arrow ([18F]F-, toluene, 105–110 °C, 10 min.) and the second arrow (aq. HCl conc., 120 °C, 7 min., ethoxymethyl).
Round 2
Reviewer 2 Report
i recommend publication